# *Zingiber officinale* Uncovered: Integrating Experimental and Computational Approaches to Antibacterial and Phytochemical Profiling

**DOI:** 10.3390/ph17111551

**Published:** 2024-11-19

**Authors:** Abdel Moneim Elhadi Sulieman, Safa Mustafa Ibrahim, Mamdouh Alshammari, Fahad Abdulaziz, Hajo Idriss, Naimah Asid H. Alanazi, Emad M. Abdallah, Arif Jamal Siddiqui, Sohair A. M. Shommo, Arshad Jamal, Riadh Badraoui

**Affiliations:** 1Department of Biology, College of Science, University of Ha’il, Ha’il 81451, Saudi Arabia; mamd.alshammari@uoh.edu.sa (M.A.); n.alenezy@uoh.edu.sa (N.A.H.A.); ar.siddiqui@uoh.edu.sa (A.J.S.); arshadjamalus@yahoo.com (A.J.); badraouir@yahoo.fr (R.B.); 2Department of Microbiology, Faculty of Sciences, University of Gezira, Wad-Medani 21111, Sudan; safa_ibrahim1978@gamil.com; 3Department of Chemistry, College of Science, University of Ha’il, Ha’il 81451, Saudi Arabia; fah.alanazi@uoh.edu.sa; 4Department of Physics, College of Science, Imam Mohammad Ibn Saud Islamic University (IMSIU), Riyadh 11432, Saudi Arabia; hiidriss@imamu.edu.sa; 5Deanship of Scientific Research, Imam Mohammad Ibn Saud Islamic University (IMSIU), P.O. Box 5701, Riyadh 11432, Saudi Arabia; 6Department of Biology, College of Science, Qassim University, Buraydah 51452, Saudi Arabia; 140208@qu.edu.sa; 7Department of Sport Science and Physical Activity, College of Education, University of Ha’il, Ha’il 81451, Saudi Arabia; susu.shommo@gmail.com; 8Section of Histology-Cytology, Medicine Faculty of Tunis, University of Tunis El Manar, La Rabta, Tunis 1007, Tunisia

**Keywords:** in vitro study, pharmacological properties, bioactive molecules, pharmacokinetics, computational biology

## Abstract

Background: *Zingiber officinale* rhizome is widely cultivated in the central region of Sudan (Gezira) and data on the biological properties of this variety grown in Sudan’s climate are scarce. This study aims to comprehensively analyze the antibacterial, antioxidant, phytochemical, and GC-MS properties of *Zingiber officinale* (ginger rhizome) to explore its potential applications. Methods and Results: The in vitro antibacterial assessment of the aqueous extract of Sudanese ginger revealed moderate activity against *Staphylococcus aureus*, *Salmonella typhi*, *Pseudomonas aeruginosa*, *Escherichia coli*, and *Klebsiella pneumonia*, as determined by the disc diffusion method. The inhibition zones ranged from 12.87 ± 0.11 mm to 14.5 ± 0.12 mm at 30 µg/disc. The minimum inhibitory concentration ranged from 6.25 to 25 µg/mL, while the MBC ranged from 25 to 50 µg/mL. The MBC/MIC exhibited a bactericidal effect against all tested bacteria. Phytochemical screening revealed the presence of various chemical constituents, such as saponins, flavonoids, glycosides, alkaloids, steroids, terpenoids, and the absence of tannins in Sudanese ginger rhizome. Furthermore, GC-MS analysis of ginger rhizome identified 22 chemical compounds with retention times ranging from 7.564 to 17.023 min. The identification of 22 chemical compounds through GC-MS analysis further underscores the prospect of harnessing ginger rhizome for the development of novel medications. Computational analyses showed that ginger compounds bind the Protein Data Bank (PDB) codes 1JIJ and 2QZW with high binding affinities, reaching −9.5 kcal/mol. Ginger compounds also established promising molecular interactions with some key residues, satisfactorily explaining the in vitro results and supporting the pharmacokinetic and experimental findings. Conclusions: This study lays the groundwork for future research and pharmaceutical exploration aimed at harnessing the beneficial properties of ginger rhizome for medicinal and therapeutic purposes, particularly its antimicrobial potential.

## 1. Introduction

Recently, there has been increasing interest in biologically active compounds obtained from natural sources. Plant extracts are utilized in the food industry as preservatives and antioxidants, in cooking as flavoring agents, in cosmetology as components in skin care products, and in traditional medicine for their antidiabetic, antibacterial, antiviral, anti-inflammatory, antimalarial, wound-healing, immunomodulatory, or hemostatic properties. In addition, certain plant extracts are utilized in the biological hemostatic manufacturing of cellulose nanofibers, silver nanoparticles, and copper and zinc oxides as reducing or stabilizing agents [1,2,3,4]. Ginger, a rhizome of the *Zingiber officinale* plant, has been used for centuries in traditional medicine for its potential health benefits, such as assisting digestion and alleviating symptoms of stomach distress, including diarrhea and nausea. Additionally, it may be utilized to treat conditions such as arthritis, colic, and heart disorders [5,6,7,8,9]. In recent years, researchers have investigated ginger’s antibacterial properties against various Gram-positive and Gram-negative bacteria [10], as well as its phytochemical profile and GC-MS analysis showing distinct compositional differences among the ginger cultivars [11].

A growing body of evidence suggests ginger has antibacterial solid properties due to its rich chemical composition. In numerous studies, ginger has been shown to inhibit the growth of many bacteria, including *Escherichia coli*, *Staphylococcus aureus*, and *Salmonella enterica*. The active compounds in ginger, such as gingerol, shogaol, and paradol, are believed to be responsible for these antibacterial effects [12,13,14]. These compounds interact with bacterial cell membranes, disrupting their integrity and ultimately leading to cell death. Additionally, ginger has been found to have a synergistic effect when combined with certain antibiotics, enhancing its efficacy against resistant strains of bacteria [15].

Ginger is rich in phytochemicals and bioactive compounds found in plants linked to numerous health benefits [16]. The rhizome contains some bioactive compounds, including flavonoids, terpenoids, and phenolic acids. It has been demonstrated that these phytochemicals have anti-inflammatory, anticancer, and antioxidant qualities, as well as anti-inflammatory and anticancer properties. Based on the Folin–Ciocalteu assay, the total polyphenol content of *Zingiber officinale* rhizome extract was measured at 181.41 mg GAE/g of extract, with flavonoids accounting for 7.8% (14.15 mg quercetin/g of extract) [17]. In addition to its antibacterial properties, ginger has been found to contain phytochemicals with a wide range of health benefits. These compounds, including flavonoids, terpenoids, and phenolic acids, have been shown to possess antioxidant, anti-inflammatory, and anticancer properties, making ginger a potential superfood for promoting overall health and well-being.

Turmeric, cardamom, and galangal are other plants in the Zingiberaceae family. These plants share similar botanical characteristics and are widely used for culinary and medicinal purposes [18]. The present study aims to comprehensively analyze ginger rhizome’s antibacterial, phytochemical, and GC-MS properties and explore its possible applications. The identified ginger phytochemicals have been assessed for their pharmacokinetic properties, binding activities, and molecular interactions with the TyrRS from *Staphylococcus aureus* and the aspartic proteinase from *Candida albicans* [19].

On a global scale, there is enormous concern about the potential decline of antibiotic efficacy due to the increasing prevalence of multi-drug-resistant bacteria. As a result, scientists are actively exploring different alternatives from many sources [20]. The WHO has published a list of critical and high-priority clinical bacteria, including drug-resistant *Acinetobacter baumannii*, *Pseudomonas aeruginosa*, *Escherichia coli*, *Klebsiella pneumoniae*, and MRSA, among others, emphasizing the need for urgent action to control the rapid spread of these pathogens [21,22].

Regrettably, there is a scarcity of comprehensive data about the pharmacological qualities of Sudanese medicinal plants. According to the World Health Organization (WHO), “The Sudan Atlas of Medicinal Plants” documents more than 2000 therapeutic plants. However, it is suspected that the current number of medicinal plants in use may be higher [23]. Plant composition is well known to vary significantly depending on climate conditions [24]. The climate in central Sudan is tropical, characterized by elevated humidity and precipitation levels.

Nevertheless, further study on *Zingiber officinale* rhizome is necessary. Therefore, this study aimed to provide a detailed insight into the various chemical constituents present in ginger rhizome cultivated in Sudan, along with their potential antibacterial effects using in vitro methods and computational assays (in silico) on Tyrosyl-tRNA Synthetase (TyrRS) from *Staphylococcus aureus* (Protein Data Bank code: 1JIJ) and aspartic proteinase from *Candida albicans* (Protein Data Bank code: 2QZW). The findings of this research contribute to the global understanding of ginger’s health benefits and support its widespread use in various medical and culinary applications.

## 2. Results

### 2.1. The Antibacterial Properties of Ginger Rhizome

The results of the antibacterial activity using the disc diffusion method are shown in Table 1. The in vitro testing revealed that Sudanese ginger aqueous extract exhibited moderate antibacterial activity against several bacteria, including *Staphylococcus aureus*, and weak activity against *Salmonella typhi* and *Klebsiella pneumoniae*. The results indicated that the mean inhibition zone at the higher concentration of the extract (30 µg/disc) averaged from 14.5 ± 0.0 mm to 12.87 ± 0.11 mm, while at the lower concentration, it averaged from 11.0 ± 0.1 mm to 6.33 ± 0.33 mm. The evaluation of the sizes of inhibition zones was conducted based on the disc diffusion test for bacteria, using the following criteria: a diameter of 10 mm or less suggests low activity, a diameter between 10 and 15 mm represents moderate activity, and a diameter beyond 15 mm reflects vigorous activity [25]. Furthermore, the data showed that *Staphylococcus aureus* was the most highly inhibited bacteria by the ginger extract. *Staphylococcus aureus* (Gram-positive) showed higher inhibition zones than other bacteria (Gram-negative) due to its thicker but more permeable peptidoglycan layer. In contrast, the Gram-negative bacteria have an outer membrane that acts as an additional barrier, reducing antibacterial penetration and effectiveness [26,27]. Our results corroborate previous studies; it was reported that the aqueous extract of ginger rhizome exhibits antibacterial activity against *Escherichia coli* and *Bacillus subtilis* [28]. It was also reported that antibacterial effects were still observed after mixing soybean oil extract and ginger extract. This combination showed acceptable activity against 24 bacterial isolates belonging to six distinct types of foodborne pathogens: *Escherichia coli*, *Pseudomonas aeruginosa*, *Staphylococcus aureus*, *Klebsiella* spp., *Vibrio cholerae*, and *Salmonella* spp. The range of inhibitory zones varied from 11.67 ± 1.53 mm to 8.0 ± 1.73 mm [29]. Interestingly, the essential oils of ginger were reported to have high antibacterial activity, as was the diameter of the zone of inhibition against *Staphylococcus aureus*, which was recorded as 17.10 mm [30].

The M.I.C. and M.B.C. results, crucial in determining the potency of ginger extract, are presented in (Table 2). These results align with the findings of the disc diffusion test. Notably, the most susceptible bacterium was *Staphylococcus aureus* (M.I.C. = 6.25 µg/mL, M.B.C. = 25 µg/mL), possibly due to the unique bioactive compounds in ginger extract that are particularly effective against cellular structures or metabolic processes. A previous study suggested that the antibacterial properties of ginger are linked to the presence of bioactive compounds [30]. *Staphylococcus aureus* recorded (M.I.C. = 6.25 µg/mL, M.B.C. = 50 µg/mL), indicating that while the extract effectively inhibits bacterial growth at lower concentrations, higher concentrations are needed for bactericidal effects. The thicker peptidoglycan layer of Gram-positive bacteria like *Staphylococcus aureus* may necessitate a higher extract concentration for complete bacterial eradication [31]. *Pseudomonas aeruginosa*, *Escherichia coli*, and *Klebsiella pneumonia* exhibited similar levels of susceptibility with an M.I.C. of 25 µg/mL and an M.B.C. of 25 µg/mL. These Gram-negative bacteria have an outer membrane that provides an extra barrier to the extract, which could explain the higher M.I.C. and M.B.C. values compared to *Staphylococcus aureus*. However, since the M.B.C. is equal to the M.I.C. for these bacteria, it suggests that the extract is equally effective at inhibiting and killing these pathogens once the threshold concentration is reached. The M.B.C./M.I.C. ratio, calculated for the time-kill assay, suggests that ginger extract has bacteriostatic effects on *Staphylococcus aureus* and bactericidal effects on the other tested bacteria. This conclusion is drawn from the fact that the effect is classified as bactericidal if the ratio of M.B.C./M.I.C. is less than or equal to 4. Conversely, if the ratio was more significant than 4, the impact was categorized as bacteriostatic [32]. Genetic variation and environmental adaptation may influence this phenomenon.

### 2.2. Phytochemical Properties Results

The provided excerpt presents the results of a phytochemical screening of ginger rhizome in Table 3. The screening revealed the presence of various compounds in the ginger rhizome, specifically flavonoids, saponins, glycosides, alkaloids, and terpenoids. Interestingly, tannins were not found in the ginger rhizome. This information indicates the diverse array of phytochemicals in these plant parts, which could have implications for their potential uses in various applications such as traditional medicine or food science.

### 2.3. GC-MS Result

The GC-MS analysis of ginger rhizome revealed the presence of 22 different compounds, with gingerol being the most abundant at 41.30% concentration. Other significant compounds identified include zingiberene (14.04%), β-d-glucopyranose, 4-*O*-beta-d-galactose (7.96%), cyclohexane, 3-(1,5-dimethyl-4-hexenyl) (6.69%), 4-hydroxybenzoic acid (4.65%), β-bisabolene (3.44%), and exo-2,7,7-trimethylbicyclo[2,2,1]heptane-2-ol (2.39%). These compounds were distributed within a retention time range of 5.106–19.351 min, providing a comprehensive chemical profile of ginger rhizome (see Table 4, Figure 1).

The dominance of gingerol in the composition of ginger rhizome is noteworthy, as it is a bioactive compound associated with various health benefits. Additionally, the presence of zingiberene, β-d-glucopyranose, 4-*O*-beta-d-galactose, cyclohexaene, 4-hydroxybenzoic acid, β-bisabolene, and exo-2,7,7-trimethylbicyclo[2,2,1]heptan-2-ol contributes to the overall chemical complexity and potential pharmacological properties of ginger. This analysis provides valuable insight into the chemical composition of ginger rhizome, offering a foundation for further research into its medicinal and culinary uses.

### 2.4. Computational Modeling Results

Table 5 shows the identified compounds’ lipophilicity, bioavailability, and pharmacokinetic properties based on their absorption, distribution, metabolism, excretion, and toxicity (ADMET) properties. Our findings showed acceptable ADMET properties for most ginger-identified compounds (Figure 2). Most of the latter did not inhibit the five assessed cytochrome P450 (C.Y.P.s) isoforms (1A2, 2C19, 2C9, 2D6, and 3A4) and possessed good oral bioavailability. Furthermore, all the ginger-identified compounds obeyed the Lipinski rule. They were predicted not to be substrates of P-glycoprotein (P-gp), which means there will be no disruption of distribution and metabolism. The synthetic accessibility varied between 1 and 4.81, which indicates that these compounds are easy to synthesize. Ginger compounds bound TyrRS from *Staphylococcus aureus* (Protein Data Bank code: 1JIJ) and aspartic proteinase from *Candida albicans* (Protein Data Bank code: 2QZW) with acceptable affinities (from −4.5 to −9.5 kcal/mol) (Table 6). All the 22 ginger-identified compounds had negative binding affinities that support their potential bioactivities. Compound no. **22** had negative affinities of −9.5 kcal/mol for 1JIJ and −7.5 kcal/mol. The compounds with the best scores (No. **22**, **20**, **16**, and **9**) were further analyzed for bond category, molecular interactions, and deep embedding, as shown in Table 7 and Figure 3. These compounds established good molecular interactions with 1JIJ and 2QZW (Table 6, Figure 4).

The molecular interactions included conventional H-bonds associated with a network of electrostatic and hydrophobic bonds. Taken together, the computational modeling analyses showed that the antimicrobial effects of ginger phytochemicals are thermodynamically possible. These antimicrobial effects had already been reported in the current study using in vitro analyses.

## 3. Discussion

The present study aims to comprehensively analyze ginger rhizome’s antibacterial, phytochemical, and GC-MS properties and explore its possible applications. This study provided a detailed insight into the various chemical constituents present in ginger rhizome, their potential antibacterial effects, and possible therapeutic uses. The findings of this research will contribute to the global understanding of ginger’s health benefits and support its widespread use in various medical and culinary applications.

### 3.1. Antibacterial Properties

The ginger extract exhibited varying levels of effectiveness against different bacteria, with M.I.C. values ranging from 6.25 to 50 µg/mL. The extract demonstrated bacteriostatic activity against *Staphylococcus aureus* and *P. aeruginosa* while displaying bactericidal action against *Klebsiella pneumoniae* and *Escherichia coli.* These findings indicate the potential of the ginger extract as a source of antibacterial agents, which can inhibit and kill certain pathogenic microorganisms. The M.B.C./M.I.C. ratio is greater than or equal to 4 for the latter two bacteria, further supporting the bactericidal nature of the extract’s action. This information contributes to the understanding of the potential antimicrobial properties of ginger and its potential applications in developing new antibacterial agents.

In previous studies, ginger has been shown to inhibit the growth of various bacteria, including *Escherichia coli*, *Staphylococcus aureus*, and *Salmonella enterica* [33]. The active compounds in ginger, such as gingerol, shogaol, and paradol, are believed to be responsible for these antibacterial effects [12,13]. These compounds interact with bacterial cell membranes, disrupting their integrity and ultimately leading to bacterial death. Additionally, ginger has been found to have a synergistic effect when combined with certain antibiotics, enhancing their efficacy against resistant strains of bacteria [15].

Numerous researchers have reached the consensus that *Zingiber officinale* exhibits exceptional antimicrobial properties. However, they attributed these properties to naphthalenamine, decanal, alpha-copaene, and numerous other chemical constituents [14,34,35]. On the other hand, Shareef et al. [36] utilized the diffusion method on agar to evaluate the in vitro antibacterial activity of a methanolic extract containing bioactive compounds from *Zingiber officinale* against *Proteus mirabilis*, *Escherichia coli*, *P. aerogenosa*, *Staphylococcus aureus*, and *Klebsiella pneumoniae*. Inhibitory zones were compared to those of various standard antibiotics. All regimens’ inhibition zone diameters varied between 4.93 ± 0.290 and 0.89 ± 0.210 mm.

Overall, the antibacterial study demonstrated that ginger aqueous extract possesses significant in vitro antibacterial activity against the tested bacteria. As indicated in the tables, the results highlighted dose dependent inhibition zones. In fact, higher concentrations generally resulted in larger inhibition zones. Moreover, the specific impact of the ginger extract on *Staphylococcus aureus*, the most highly inhibited bacteria, underscores its potential as an effective antibacterial agent. These findings contribute to understanding ginger’s medicinal properties and its potential application in combating bacterial infections.

### 3.2. Phytochemical Properties

Phytochemical analysis showed that Sudanese ginger rhizome revealed the presence of various compounds. Our findings were similar to other investigators who found that ginger is also rich in phytochemicals, which are bioactive compounds in plants associated with numerous health benefits. Our results revealed the presence of many bioactive compounds, including flavonoids, terpenoids, and phenolic acids. These phytochemicals have been shown to possess antioxidant, anti-inflammatory, and anticancer properties, among others. In particular, ginger flavonoids, such as quercetin and kaempferol, have been extensively studied for their potential to combat various diseases [37,38]. Among the phenolic chemicals found in ginger, studies have shown that zingerone, gingerols, and shogaols are the most common. Of these, gingerols are commonly found in greater abundance in fresh ginger, while shogaols are formed through heat treatment. Zingiberene, β-bisabolene, and sesquiphellandrene are common terpenoids that are part of another important group of bioactive compounds. These compounds have antimicrobial and anti-inflammatory effects in various varieties, including those from Nigeria, China, and India [39,40]. Research conducted in different climates on these chemicals has shown that, although all rhizomes of *Z. officinale* contain the same class of compounds, the amounts and bioactivity of these compounds might vary [39,41].

By comparing the GC-MS results of Sudanese ginger with those from a published study by Elbestawy et al. [42] on Egyptian ginger, our study identified a total of 22 compounds in Sudanese ginger rhizome, whereas the published study identified 17 distinct compounds. Both studies revealed similarities and differences in the chemical composition of ginger rhizome. Gingerol and zingiberene were identified as the primary compounds in both, albeit in slightly different concentrations. Additional shared compounds include α-terpineol and β-bisabolene. Both studies emphasize ginger’s pharmacological potential, noting its anti-inflammatory, antimicrobial, and antioxidant effects. However, there are notable differences. The published study detected thymol at a concentration of 10.50%, a compound absent in our analysis. Thymol is known for its antimicrobial and antifungal properties, contributing an additional aspect to ginger’s medicinal profile. Conversely, our study identified unique compounds such as β-d-glucopyranose, 4-hydroxybenzoic acid, and exo-2,7,7-trimethylbicyclo[2.2.1]heptane-2-ol. Additionally, the published study identified shogaol and α-bergamotene, unique compounds with anti-inflammatory properties that further enhance ginger’s therapeutic potential. These differences highlight potential variations in ginger’s chemical composition, influenced by factors such as plant origin, extraction methods, and analytical conditions. We conclude that the constituents of ginger are significantly affected by the cultivation environment. Our observation is supported by a previous study on ginger from Indonesia and Japan, where GC-MS analysis showed clear differences in composition between the ginger cultivars [11].

Moreover, Thakor et al. [43] found that the qualitative examination of phytochemicals revealed the presence of saponins, alkaloids, flavonoids, and steroids in ginger extracts. Ginger in acetone had the best antibacterial activity against *Escherichia coli* MTCC 334 with 24 mm of clear zone, whereas ginger in methanol had the lowest activity against *Bacillus subtilis* MTCC 441 with 10 mm of clear zone.

The presence of phenols and tannins, alkaloids, flavonoids, terpenoids, sterols, cardiac glycosides, and saponins in the extracts has various implications across different fields. For example, phenols and tannins are known for their antioxidant properties and are often associated with potential health benefits. Alkaloids are present in many medicinal plants and have various pharmacological effects. They belong to a class of naturally occurring compounds with various pharmacological properties. Plant-based natural compounds, such as alkaloids, have shown promise as preventive measures against chronic inflammation and neurodegenerative diseases (N.D.D.s) [38].

Antioxidant and anti-inflammatory qualities are well-known for flavonoids [9,10]. Many health advantages are associated with flavonoids, such as their antiviral, anticancer, and antioxidant qualities. They also have cardio- and neuroprotective properties.

The potential of terpenoids as antibacterial and anticancer agents has been investigated. In the pharmaceutical business, steroids are crucial for the synthesis of medications like corticosteroids [44]. For many years, cardiac glycosides have been prescribed as medications to treat arrhythmias and heart failure. The principal mechanism of action of these substances is their inhibition of Na^+^/K^+^-ATPase (NKA), which controls the intracellular concentration of calcium ions and sodium and potassium ions. These changes in the intracellular concentration of Ca^2+^ result in a positive inotropic effect on the heart muscle, for which they are used in the indications above [45,46].

Research has been conducted on saponins’ potential as antifungal and anticancer drugs. Consequently, the presence of these chemicals in the extract points to possible industrial, medical, and pharmacological uses. High structural diversity in saponins is associated with their anticancer properties. Numerous studies have demonstrated the role of saponins in cancer and their mechanism of action, including antioxidant activity, suppression of cellular invasion, cell-cycle arrest, induction of autophagy, and apoptosis. There are currently no known FDA-approved saponin-based anticancer medications, despite the substantial anticancer properties of saponins and the wealth of research on them. Numerous restrictions, such as toxicities and drug-like qualities, can be blamed for this [47]. The antibacterial properties of Sudanese ginger are linked to its phytochemical composition, which can be influenced by environmental factors like climate. Temperature, sunlight, and rainfall are crucial in this regard, with plants in warmer climates typically producing higher levels of secondary metabolites [48].

This brief passage highlights the importance of phytochemical screening in identifying the chemical composition of plant materials. Overall, ginger rhizomes exhibit a diverse array of bioactive compounds, which could have implications for their potential uses in various applications, such as traditional medicine or food science.

### 3.3. The GC-MS Constituents

The GC-MS analysis of Sudanese ginger rhizome identified 22 various compounds, including two terpenoids (zingiberene and β-bisabolene), three *N*-containing compounds (Benzeneethanamine, 2,5-dimethoxy-, 2,4-bis(hydroxyamino)-5-nitropyrimidine, and 1-(5-bicyclo[2,2,1]heptyl)ethylamine), one phenolic compound (Phenol, 2-methoxy-3-(2-propenyl)-), and two esters (Docosanoic acid, ethyl ester, and E-11-hexadecenoic acid, ethyl ester). The concentrations of the main constituents were as follows: gingerol at 41.30%, zingiberene at 14.04%, β-d-glucopyranose, 4-*O*-beta-d-galactose at 7.96%, cyclohexane, 3-(1,5-dimethyl-4-hexenyl)- at 6.69%, 4-Hydroxybenzoic acid at 4.65%, β-bisabolene at 3.44%, and Exo-2,7,7-trimethylbicyclo[2,2,1]heptane-2-ol at 2.39%.

The GC-MS analysis of ginger rhizome did not detect any Cl-containing compounds. The GC-MS analysis of ginger rhizome identified several key compounds, with gingerol being the most abundant. Various other compounds within the rhizome further enhance its potential for pharmaceutical and therapeutic applications. This comprehensive chemical profiling is essential for understanding ginger’s biological activities and potential health benefits, paving the way for its utilization in various fields, including medicine, nutrition, and natural product development. Compared to earlier research, ginger essential oil had 45 phytochemical components, with Geranial, 1,8-Cineole, Neral, Camphene, α-zingiberene, and α-Farnesene being the most abundant [49].

As a frequent condiment or essential spice in food and drink, ginger is widely used. Zingerone, shogaols, gingerols, paradols, wikstromol, and carinol were found to be the main constituents of the pungent compounds, according to GC-MS analysis [50,51]. By steam distillation, the volatile chemicals that give ginger its flavor have been removed. According to the data, more than 90 components were discovered and separated. The most prevalent chemical found was zinziberene. According to Jedli et al. [16], chemical investigations on ginger-based dietary supplements showed the presence of gingerols, shogaols, parasols, and gingerdiones.

Dafaalla [52] identified twenty-two distinct components from ginger rhizome’s hexane extraction, and the primary constituent was the alkaloid gingerol (18%).

Furthermore, the GC-MS analysis revealed a diverse array of chemical compounds present in the Argel leaves. The compounds identified belong to various chemical classes, including alcohols, ketones, aldehydes, furans, acids, esters, etc. Additionally, the area percentages of the identified compounds provide insights into their relative abundance within the sample. For instance, the relatively high area percentage of 3-Pentanol,2,2,4,4-tetramethyl- suggests its significant presence in the argel leaves, while compounds with lower area percentages may be in smaller quantities. The comprehensive nature of the analysis and the diverse range of identified compounds underscore the complex chemical composition of argel leaves, shedding light on their potential pharmacological and therapeutic properties.

### 3.4. Computational Modeling

Table 7 shows the lipophilicity, bioavailability, and pharmacokinetic properties of the identified compounds based on the absorption, distribution, metabolism, excretion, and toxicity (ADMET) properties. Our findings showed acceptable ADMET properties for most of the ginger-identified compounds. Most of the latter did not inhibit the five assessed cytochrome P450 (C.Y.P.s) isoforms (1A2, 2C19, 2C9, 2D6, and 3A4) and possessed good oral bioavailability (Table 7). These predictions have been confirmed by the mapped boiled egg model (Figure 2) and supported the high gastrointestinal (G.I.) absorption and the blood–brain barrier (B.B.B.) permeation. Skin absorption varied between 0 and 136, which means low to high permeation [53,54]. Furthermore, all the ginger-identified compounds obeyed the Lipinski rule. They were predicted not to be P-glycoprotein (P-gp) substrates, which means there will be no disruption of distribution and metabolism [53,54]. The synthetic accessibility varied between 1 and 4.81, which indicates that these compounds are easy to synthesize [53,54].

Ginger compounds bound TyrRS (Tyrosyl-tRNA Synthetase) from *Staphylococcus aureus* (1JIJ) and aspartic proteinase from *Candida albicans* (2QZW) with acceptable affinities (from −4.5 to −9.5 kcal/mol) (Table 5). Recently, it has been shown that binding affinities depend mainly on the ligands’ 3D chemical structure and structural geometry [55,56]. All the 22 ginger-identified compounds had negative binding affinities that support their potential bioactivities. The compound no. **22** had negative affinities of −9.5 kcal/mol for 1JIJ and −7.5 kcal/mol. The compounds with the best scores (No. **22**, **20**, **16**, and **9**) were further analyzed for bond category, molecular interactions, and deep embedding (see Table 7). These compounds established good molecular interactions with 1JIJ and 2QZW (Table 6).

The molecular interactions included conventional H-bonds associated with a network of electrostatic and hydrophobic bonds. As previously described, this network contributes to the stability of the ligand-receptor complex [57,58]. These interactions involved several key residues and deep embedding (<2.5 Å). Tight embedding is commonly associated with potential biological effects, including anti-inflammatory, antiproliferative, antioxidant, and antimicrobial effects [53,54]. Previous studies outlined several benefits of ginger compounds. Nevertheless, our GC-MS phytochemical composition, as reported in Table 4, is different and might have particular outcomes. In fact, such a combination might result in particular antimicrobial findings. In this context, beneficial effects can be greater than the sum of the individual effect of each ginger-identified compound [12,55]. Taken together, the computational modeling analyses showed that the antimicrobial effects of ginger phytochemicals are thermodynamically possible. These antimicrobial effects had already been reported in the current study using in vitro analyses. These results confirmed the health promotion and promising benefits of natural-derived compounds and their phytotherapeutic potential, especially the ginger rhizome.

## 4. Materials and Methods

### 4.1. Chemicals

All the chemicals employed in the study were of analytical grade. These chemicals and indicators were utilized in their original state without any purification processes, and were procured from Merck (Merck^®^, Rahway, NJ, USA).

### 4.2. Samples Collection and Preparation

The ginger rhizomes of (*Zingiber officinale*), as shown in Figure 5, used in this investigation were obtained from the local markets of Wad-Medani town (located in Central Sudan) on 15 April 2022 at the geographical coordinates Latitude: 14.39795320, Longitude: 33.52513160, and Elevation: 411.46 m. Afterwards, the specimens were sent to the Central Laboratories of the University of Gezira. The identification technique followed a rigorous and systematic approach, which included conducting morphological analysis utilizing identification keys and referring to herbarium records from the Gezira University Herbarium. Dr. Mutaman Ali Abdelgadir Kehail from the College of Sciences identified the rhizomes and assigned them a voucher number: Keh-15APR2022-GeziraUniv-Sudan. The rhizomes were washed with flowing distilled water, left to air-dry at room temperature for 48 h, and then crushed into fine powder using a blender. For ethanolic extraction, 100 g of finely powdered ginger rhizomes were placed in an orbital shaker incubator at room temperature. They were shaken regularly in 1000 mL of 80% ethanol for several days. The ethanolic extract was concentrated using a rotary evaporator (8 kW, 50 L, Henan Lanphan Industry Co., Ltd, Zhengzhou, China). The obtained solution was then filtered through filter paper, and the extract was dried on a glass Petri dish at room temperature. The powdered ginger extract was used for phytochemical screening and GC-MS analysis.

### 4.3. GC-MS Analysis

A conventional analytical method suggested by Banso and Adeyemo in [59] was used for qualitative phytochemical analysis to determine for the presence of phenols and tannins, alkaloids, flavonoids, terpenoids, sterols, cardiac glycosides, and saponins in ethanol and aqueous ginger rhizome extracts. These methods are established protocols for chemical analysis and are widely recognized in the scientific community for their reliability and accuracy in identifying the presence of specific compounds in extracts. By following these methods, researchers can ensure the credibility and reproducibility of their findings, as well as gain valuable insights into the potential uses and properties of the analyzed compounds. The compounds analyzed include a wide range of chemical classes, each with its own significance in various fields such as pharmaceuticals, agriculture, and environmental studies. This comprehensive approach to chemical analysis aims to identify the presence of key compounds in the extracts, providing valuable insights into their potential uses and properties [57,58]. For qualitative phytochemical analysis, 0.5 g of ginger rhizomes extracts was diluted in 100 mL each of ethanol and water to provide a stock solution concentration of 5 mg/mL.

First, 0.1 g of the extract was dissolved in 10 mL of analytical-grade ethanol. The solution was then passed through a 0.45 mm syringe filter and into a 1.5 mL vial, where it was prepared for injection into GC-MS. The samples were analyzed using GC-MS at the Central Laboratories of the University of Gezira. The GC-MS analysis identified various chemical constituents present in the extract, providing detailed information such as their retention time, base peak, molecular weight, molecular formula, and compound names. The NIST 14S library was utilized to identify the compounds detected in the samples.

The conditions of analysis are as follows:The split ratio of 1 L injected sample was 10:1. The oven temperature program was set to 280 °C for 25 min at 80 °C/minute from 60 °C. The show was 53.5 min. Conditions for GC-MS leaf oil analysis: as noted, GC-MS detected FAME molecules.The helium flow was 0.7 mL/min. The ion source, transfer, and injector were heated to 250, 250, and 220 °C. After 1 min at 50 °C, the oven was heated to 250 °C at a rate of 40 °C each minute. From 35 to 500 amu, full-scan mass spectra were acquired for all data. Spectra were compared to mass spectral databases to identify substances. We set system calibration and minimal detection limits using manufacturing circumstances. Other sources provide the equation [30].

This analytical approach allowed for the comprehensive characterization of the chemical composition of the ginger rhizome extract. The study’s findings offer valuable information regarding the chemical profile of the ginger rhizome extract, laying the groundwork for potential applications in various fields such as pharmacology, food science, and natural product research.

### 4.4. Bacterial Strains

A range of five pathogenic bacteria including one Gram-positive (*Staphylococcus aureus*) and four Gram-negative (*Escherichia coli*, *Klebsiella pneumoniae*, *Pseudomonas aeruginosa*, and *Salmonella typhi*) were generously provided by the Department of Microbiology of the University of Gezira. The source of bacterial is clinical isolates previously identified and sent from Wad Madani Hospital, Sudan. These strains were chosen for their clinical significance as common pathogens, making them ideal for evaluating the antibacterial efficacy of ginger. The pure bacterial samples were subcultured for further research by growing the bacteria on Brain Heart Infusion Agar (B.H.I. agar) for overnight incubation at 35–37 °C. Before the experiment, a loop-full of the bacterial specimen was adjusted to the McFarland standard, using sterile normal saline (0.9%) to make the working bacterial solution, which equals approximately 10^6^ CFU/mL.

### 4.5. Preparation of the Plant Extract

In aqueous extraction, 20 g of air-dried powder was added to 150 milliliters of distilled water and gently boiled for 2 h. The fluid was filtered through eight layers of muslin fabric and centrifuged at 5000 times gravity for 10 min. The liquid supernatant was carefully collected. The technique was repeated two times. After 6 h, the supernatant was collected at 2 h intervals, consolidated, and concentrated to one-fourth of the starting volume. This systematic approach ensures the extraction process is conducted with accuracy and attention to detail. This process is essential for achieving a thorough extraction of the desired compounds from the powder. Additionally, the gentle boiling helps to ensure that the extraction is conducted at an optimal temperature without causing degradation of the extracted compounds.

### 4.6. Antibacterial Activity Test

In the present study, the standard disc diffusion test was utilized, following the methodology described by Tambe et al. [60] with some modifications. Briefly, B.H.I. agar plates were prepared for bacterial strains, and bacteria were inoculated using the spread plate technique in a sterile environment. The inoculation was conducted using a working solution adjusted to the McFarland standard of 10^6^ CFU/mL. Discs of filter paper, with a diameter of 6 mm and made from Whatman’s No. 1 filter paper, were created and sterilized. The ginger aqueous extracts under investigation were prepared at different concentrations of 1, 2, and 3 mg/mL. Each disc absorbed precisely 10 µL of the solution, resulting in loaded discs containing 10, 20, and 30 µg of the plant extract, respectively. The filled discs were allowed to settle for a maximum of 10 min before being carefully placed on the agar surface using sterilized forceps. Subsequently, all plates were placed in an incubator and maintained at a temperature range of 35–37 °C for a duration of 24 h. After the incubation period, the diameter of the inhibition zones was measured, and the meaning of the three replicates was computed.

### 4.7. Minimum Inhibitory Concentration Test

The M.I.C. was determined using the micro-dilution method in Mueller Hinton (MH) broth as previously recommended [24,61] with minor modifications. Serial two-fold dilutions of the aqueous ginger rhizome extract (ranging from 100 to 3.125 µg/mL) were prepared in duplicate wells of a 96-well plate. Control wells containing only MH broth, as well as others containing bacteria but no extract, were also included. A 75 μL bacterial suspension in MH broth, adjusted to 10^6^ CFU/mL, was added to the designated wells. Subsequently, 75 μL of each serially diluted extract was added vertically into each well, starting from the first row. To ensure reproducibility, the entire procedure was repeated on another 96-well plate. The plates were then incubated at 35–37 °C for 24 h. After incubation, a 0.01% solution of 2,3,5-triphenyl tetrazolium chloride (T.T.C.) was added to the wells. Following an additional hour of incubation, the color change of the tetrazolium salt was observed. A lack of color change indicates growth inhibition by the biologically active extract, with the minimum inhibitory concentration (M.I.C.) resulting in complete growth suppression and no discernible color change.

### 4.8. Minimum Bactericidal Concentration Test

The minimum bactericidal concentration (MBC) of the aqueous extract derived from the rhizome of Sudanese ginger was determined against the selected bacterial strains in the following method: 100 µL from each MIC tube with no apparent growth was subcultured onto Mueller Hinton agar plates and incubated at 35–37 °C for 24 h. Subsequently, the plates underwent examination to identify any bacterial colonies. The minimum bactericidal concentration (MBC) was determined as the concentration at which no bacterial colonies were detected [25,62].

### 4.9. Phytochemical Analysis

The methods suggested by Banso and Adeyemo [59] were used to analyze the extracts ginger rhizome for the presence of phenols and tannins, alkaloids, flavonoids, terpenoids, sterols, cardiac glycosides, and saponins. These methods are established protocols for chemical analysis and are widely recognized in the scientific community for their reliability and accuracy in identifying the presence of specific compounds in extracts. By following these methods, researchers can ensure the credibility and reproducibility of their findings, as well as gain valuable insights into the potential uses and properties of the analyzed compounds.

The compounds analyzed include a wide range of chemical classes, each with its own significance in various fields such as pharmaceuticals, agriculture, and environmental studies. This comprehensive approach to chemical analysis aims to identify the presence of key compounds in the extracts, providing valuable insights into their potential uses and properties [11,42].

### 4.10. Computational Study

The pharmacokinetic properties of the ginger-identified compounds have been explored as previously reported [9,51]. The assessment of these parameters was based on the ADMET (for absorption, distribution, metabolism, excretion, and toxicity) properties [16,51].

The potential antimicrobial effect was also assessed through computational modeling and interaction assays. TyrRS (Tyrosyl-tRNA Synthetase) from *Staphylococcus aureus* (Protein Data Bank code: 1JIJ) and the aspartic proteinase from *Candida albicans* (Protein Data Bank code: 2QZW) were retrieved from the RCSB data bank. Their active sites were targeted to assess the antimicrobial effects. ChemDraw was used to draw the structure of the identified chemicals, which did not exist on the PubChem website. The CHARMm force field was applied between ligands and receptors as previously reported [59,60] following the removal of water molecules and the addition of Kollman charges and polar hydrogens. Bond types and binding scores were analyzed as previously reported [49,55]. The major reason for targeting 1JIJ and 2QZW is their implication in the pathogenesis of infectious diseases [34].

### 4.11. Statistical Analysis

The obtained data were subjected to simple descriptive analysis. The significant cases were determined through the least significant difference (L.S.D.) analysis, which was reflected by letters for each case (the different letters reflected different significant levels).

## 5. Conclusions

This comprehensive study focuses on the *Zingiber officinale* rhizome grown in the central part of Sudan (Gezira), an area that has received less attention regarding its tropical climate. The aqueous extract exhibits moderate antibacterial properties against a range of pathogens. Further investigation is recommended to focus on non-polar or semi-polar phytochemicals using different extraction methods. This suggestion is based on the results of the phytochemical screening, which identified approximately 22 bioactive chemical compounds belonging to various phytochemical classes, such as saponins, flavonoids, glycosides, alkaloids, steroids, and terpenoids. Furthermore, computational investigations revealed that ginger compounds exhibit strong binding affinities to proteins 1JIJ and 2QZW, with values as low as −9.5 kcal/mol, further confirming their potential as therapeutic agents. This work enhances the current understanding of the antibacterial and phytochemical features of ginger rhizome from Sudan and establishes a solid basis for future research. It is recommended that additional research pathways be explored in the future. It is advisable to carry out in vivo research to confirm the antibacterial activity.

## Figures and Tables

**Figure 1 pharmaceuticals-17-01551-f001:**
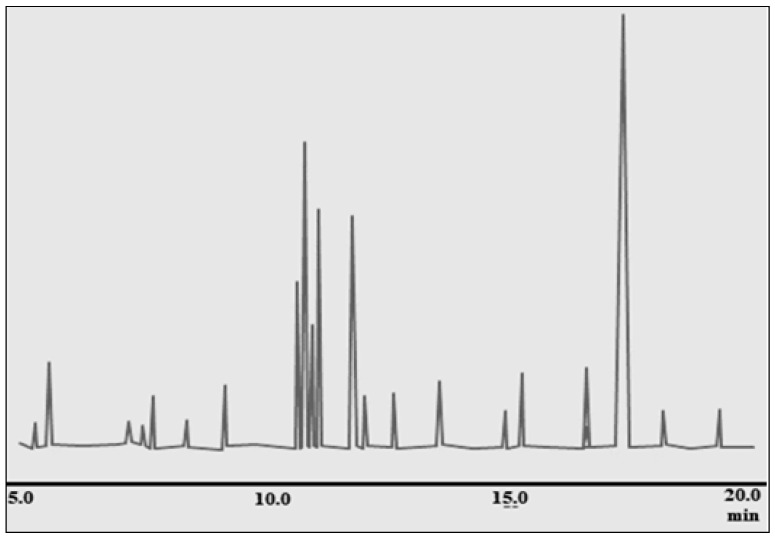
GC-MS chromatogram of ginger (*Z. officinale*) rhizome.

**Figure 2 pharmaceuticals-17-01551-f002:**
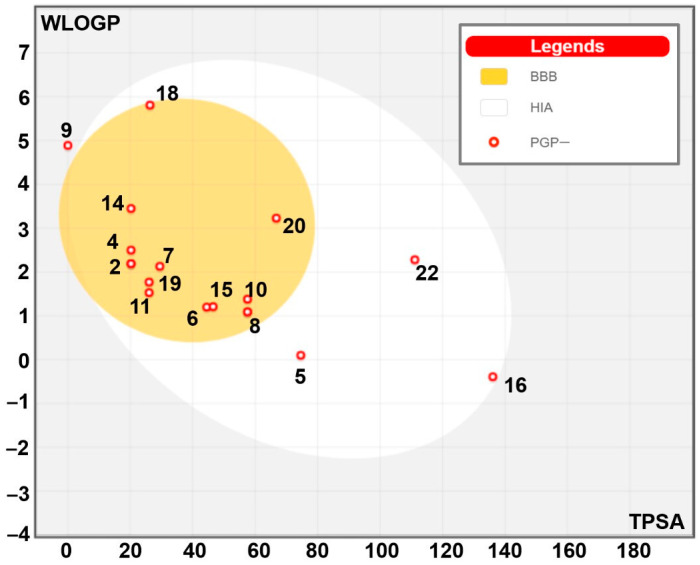
Boiled-egg model of the identified ginger phytochemicals. The yellow and white areas correspond to the BBB (blood–brain barrier) permeation and GI (gastro-intestinal) absorption, respectively.

**Figure 3 pharmaceuticals-17-01551-f003:**
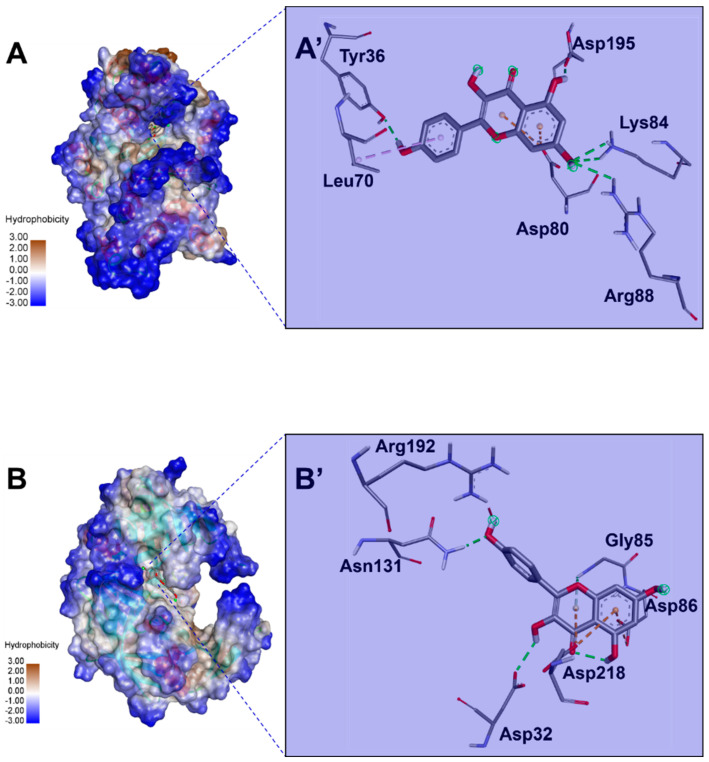
Hydrophobic illustration of compound no. **22** (kaempferol), which has the best binding affinities of −9.5 and −7.5 kcal/mol, bound to 1JIJ (**A**) and 2QZW (**B**) and their resulting 3D interactions (**A’** and **B’**, respectively). Both 1JIJ and 2QZW are Protein Data Bank codes.

**Figure 4 pharmaceuticals-17-01551-f004:**
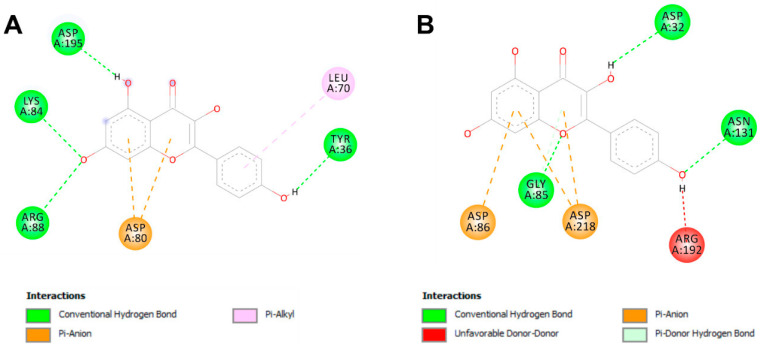
Two-dimensional diagrams of interactions of compound no. **22** (kaempferol) with 1JIJ (**A**) and 2QZW (**B**).

**Figure 5 pharmaceuticals-17-01551-f005:**
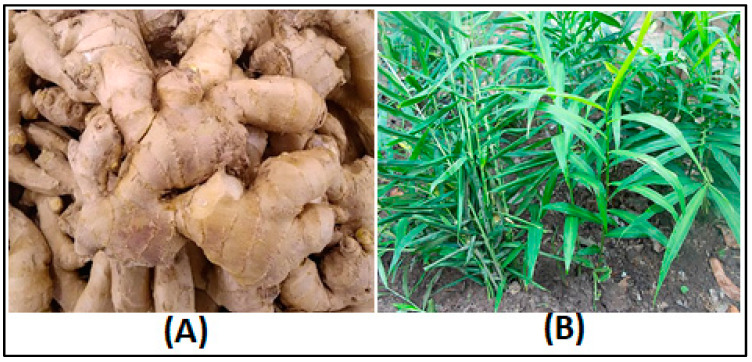
Ginger rhizome cultivated in Sudan: (**A**) the underground stem (rhizome); (**B**) the areal parts of the plant (*Zingiber officinale*).

**Table 1 pharmaceuticals-17-01551-t001:** The inhibition zones * (in mm) of ginger rhizome aqueous extract expressed as means of three replicates (mean ± SE).

Microorganism	10 µg/disc	20 µg/disc	30 µg/disc	ZOI of Ampicillin (10 µg/disc)
*Staphylococcus aureus*	11 ± 0.10 ^a^	13.0 ± 0.10 ^a^	14.5 ± 0.12 ^a^	7.33 ± 0. 23
*Salmonella typhi*	10.13 ± 0.23 ^b^	11.17 ± 0.12 ^c^	13.33 ± 0.10 ^b^	8.0 ± 1.0
*Pseudomonas aeruginosa*	7.0 ± 0.10 ^c^	11.86 ± 0.10 ^b^	13.1 ± 0.11 ^bc^	6.0 ± 0.01
*Escherichia coli*	10.2 ± 1.0 ^b^	12.67 ± 0.57 ^a^	13.33 ± 0.10 ^b^	10.12 ± 0.01
*Klebsiella pneumoniae*	6.33 ± 0.33 ^d^	10.33 ± 0.6 ^d^	12.87 ± 0.11 ^c^	9.33 ± 0.1

* Different letters reflect different significant levels with respect to the mean ± standard error. ZOI: zone of inhibition; mm: millimeter.

**Table 2 pharmaceuticals-17-01551-t002:** Determination of M.I.C.s, M.B.C.s, and M.B.C./M.I.C. ratios of tested ginger extract against the selected microorganisms expressed in µg/mL.

Microorganism	M.I.C. (µg/mL)	M.B.C.(µg/mL)	Ratio of M.B.C./M.I.C. *
*Staphylococcus aureus*	6.25	50	8
*S. typhyi*	6.25	25	4
*P. aeruginosa*	25	25	1
*Escherichia coli*	25	25	1
*Klebsiella pneumoniae*	25	25	1

* M.I.C.: Minimum inhibitroy concentration, M.B.C.: Minimum bactericidal concentration.

**Table 3 pharmaceuticals-17-01551-t003:** Phytochemical analysis of Sudanese ginger (*Z. officinale*) rhizome.

Main Class	Ginger Rhizome
Saponins	+
Flavonoids	+
Tannins	−
Glycosides	+
Alkaloids	+
Steroids	+
Terpenoids	+

(−) means absence of the main class; (+) means presence of the main class.

**Table 4 pharmaceuticals-17-01551-t004:** GC-MS of ginger (*Z. officinale*) rhizome.

Peak	Retention Time	Area %	Compound Name	Molecular Formula
1	5.106	0.69	Dihydrocarvyl acetate	C_12_H_20_O_2_
2	5.421	2.39	Exo-2,7,7-trimethylbicyclo[2,2,1]heptan-2-ol	C_10_H_18_O
3	7.079	1.62	Isoborneol	C_10_H_18_O
4	7.335	0.57	Alpha-terpineol	C_10_H_18_O
5	7.516	1.41	Butanedioic acid, 2,3-bis(acetyloxy)-, [R-(R*,R*)]-	C_8_H_10_O_8_
6	8.237	0.76	Benzeneethanamine, 2,5-dimethoxy-	C_11_H_17_NO_2_
7	9.075	1.74	Phenol, 2-methoxy-3-(2-propenyl)-	C_10_H_12_O_2_
8	10.546	4.65	4-Hydroxybenzoic acid	C_7_H_6_O_3_
9	10.711	14.04	Zingiberene	C_15_H_24_
10	10.860	3.44	β-Bisabolene	C_15_H_24_
11	11.001	6.69	Cyclohexaene, 3-(1,5-dimethyl-4-hexenyl)-	C_15_H_24_
12	11.704	7.96	β-d-Glucopyranose, 4-*O*-beta-d-galactose	C_12_H_22_O_11_
13	11.973	1.44	Hydroxycinnamic acid	C_9_H_8_O_4_
14	12.554	1.51	6,10-Dodecadien-1-yn-3-ol, 3,7,11-trimethyl-	C_15_H_24_O
15	13.503	1.87	Vanillin	C_8_H_8_O_4_
16	14.894	1.02	2,4-Bis(hydroxyamino)-5-nitropyrimidine	C_6_H_6_N_2_O_2_S_2_
17	15.201	2.07	Docosanoic acid, ethyl ester	C_24_H_34_O_2_
18	16.584	2.22	E-11-Hexadecenoic acid, ethyl ester	C_18_H_34_O_2_
19	16.892	0.57	1-(5-Bicyclo[2,2,1]heptyl)ethylamine	C_9_H_17_N
20	17.309	41.30	Gingerol	C_17_H_26_O_4_
21	18.130	0.99	Salicylic acid	C_7_H_6_O_3_
22	19.351	1.05	Kaempferol	C_15_H_10_O_6_
Total		100		

**Table 5 pharmaceuticals-17-01551-t005:** Lipophilicity, pharmacokinetics, drug-likeness and medicinal chemistry of the identified compounds based on the ADME/Tox (for absorption, distribution, metabolism, excretion, and toxicity) properties.

Entry	2	3	4	5	6	7	8	9	11	13	14	15	16	18	19	20	21	22
	Lipophilicity and Physicochemical Properties																	
TPSA	20.23	20.23	20.23	74.6	44.48	29.46	57.53	0	26.02	57.53	20.23	46.53	136.12	26.3	26.02	66.76	57.53	111.13
Log *P*o/w (iLOGP)	2.27	2.27	2.09	0.48	2.17	2.37	0.85	3.63	1.96	0.95	3.29	1.57	0.44	4.64	2.25	3.48	1.13	1.7
Consensus Log *P*o/w	2.24	2.38	2.49	-0.05	1.43	2.25	1.05	4.46	1.65	1.26	3.46	1.2	-0.98	5.61	1.96	3.13	1.24	1.58
Log S (ESOL) solubility	−2.37	−2.51	−2.87	−0.35	−1.53	−2.46	−2.07	−4.1	−1.52	−2.02	−2.87	−1.82	−1.29	−4.97	−1.91	−2.96	−2.5	−3.31
	**Pharmacokinetics**																	
GI absorption	High	High	High	High	High	High	High	Low	High	High	High	High	High	High	High	High	High	High
BBB permeant	Yes	Yes	Yes	No	Yes	Yes	Yes	No	Yes	Yes	Yes	Yes	No	Yes	Yes	Yes	Yes	No
P-gp substrate	No	No	No	No	No	No	No	No	No	No	No	No	No	No	No	No	No	No
CYP1A2	No	No	No	No	No	Yes	No	No	No	No	Yes	No	No	Yes	No	Yes	No	Yes
CYP2C19	No	No	No	No	No	No	No	Yes	No	No	No	No	No	No	No	No	No	No
CYP2C9	No	No	No	No	No	No	No	Yes	No	No	No	No	No	No	No	No	No	No
CYP2D6	No	No	No	No	No	No	No	No	No	No	No	No	No	No	No	Yes	No	Yes
CYP3A4	No	No	No	No	No	No	No	No	No	No	No	No	No	No	No	No	No	Yes
Log Kp (skin permeation)	−5.31	−5.31	−4.83	−7.2	−6.86	−5.69	−6.02	−3.88	−5.81	−6.26	−4.67	−6.37	−7.27	−3.1	−5.72	−6.14	−5.54	−6.7
	**Drug-likeness and Medicinal chemistry**																	
Lipinski	Yes	Yes	Yes	Yes	Yes	Yes	Yes	Yes	Yes	Yes	Yes	Yes	Yes	Yes	Yes	Yes	Yes	Yes
Biovailability score	0.55	0.55	0.55	0.85	0.55	0.55	0.85	0.55	0.55	0.85	0.55	0.55	0.55	0.55	0.55	0.55	0.85	0.55
Leadlikeness	1	1	1	1	1	1	1	2	1	1	3	1	1	2	1	1	1	0
Synthetic accessibility	2.35	3.43	3.24	2	1.46	1.58	1	4.81	1.24	1.61	2.96	1.15	2.66	3.11	3.3	2.81	1	3.14

TPSA, Topological polar surface area; GI, gastro-intestinal; BBB, blood–brain barrier; P-gp, P-glycoprotein; CYP, cytochrome P450.

**Table 6 pharmaceuticals-17-01551-t006:** Binding energy of ginger-identified compounds with 1JIJ and 2QZW for TyrRS from *Staphylococcus aureus* and aspartic proteinase from *Candida albicans*, respectively.

Receptor/ Ligand	1JIJ		2QZW	
	Binding Energy (kcal/mol)	RMSD (Lower–Upper)	Binding Energy (kcal/mol)	RMSD (Lower–Upper)
1	−6.0	0.0–29.5	−5.8	0.0–28.6
2	−6.1	0.0–39.1	−5.0	0.0–45.4
3	−5.5	0.0–40.7	−4.7	0.0–51.2
4	−6.5	0.0–25.7	−5.5	0.0–16.9
5	−5.9	0.0–24.6	−4.5	0.0–33.8
6	−5.7	0.0–24.1	−5.4	0.0–16.9
7	−6.4	0.0–24.9	−5.4	0.0–23.9
8	−6.6	0.0–25.0	−5.0	0.0–23.4
9	−6.9	0.0–39.4	−6.2	0.0–28.7
10	−5.9	0.0–26.7	−5.8	0.0–25.2
11	−5.0	0.0–25.7	−4.8	0.0–24.1
12	−5.8	0.0–21.6	−5.3	0.0–25.8
13	−6.4	0.0–27.4	−6.0	0.0–16.9
14	−5.4	0.0–23.8	−5.1	0.0–7.6
15	−6.2	0.0–26.5	−4.9	0.0–24.5
16	−8.1	0.0–5.6	−5.9	0.0–39.3
17	−5.3	0.0–19.6	−5.6	0.0–24.6
18	−4.8	0.0–37.3	−4.9	0.0–26.3
19	−5.6	0.0–27.7	−4.6	0.0–27.0
20	−7.1	0.0–23.7	−6.3	0.0–9.6
21	−6.7	0.0–27.7	−5.3	0.0–34.6
22	−9.5	0.0–26.3	−7.5	0.0–7.9

**Table 7 pharmaceuticals-17-01551-t007:** Molecular interactions, bond category, and closest interacting residues for the best identified compounds with the two targeted receptors: 1JIJ and 2QZW for TyrRS from *Staphylococcus aureus* and aspartic proteinase from *Candia albicans*, respectively.

Compound No.	Closest Interacting Residues		
	No. Interacting Residues	Bond Type—Residues	Closest Interacting Residue (Distance, Å)
TyrRS from *Staphylococcus aureus* (pdb id: 1JIJ)			
**22**	6	Conventional H-Bond: LYS84, LYS84, ARG88, TYR36, ASP195 Pi-Anion: ASP80, ASP80 Pi-Alkyl: LEU70	ASP195 (1.888)
**16**	5	Conventional H-Bond: ASN124, ASP40, ASP80, GLN196, ASP177, ASP40	ASP40 (2.254)
**20**	5	Conventional H-Bond: ARG58, PHE273, GLU302 Carbon H-Bond: PHE271 Pi-Pi Stacked: PHE306 Pi-Pi T-shaped: PHE273 Pi-Alkyl: PHE306, PHE306	ARG58 (1.888)
Aspartic proteinase from *Candia albicans* (pdb id: 2QZW)			
**22**	5	Conventional H-Bond: GLY85, ASN131, ASP32 Pi-Anion: ASP86, ASP218, ASP218 Pi-Donor H-Bond: GLY85	ASN131 (2.140)
**20**	5	Conventional H-Bond: ASP86, ARG192, ARG192, ARG192 Alkyl: ILE119, ILE123 Pi-Alkyl: TYR84, TYR84	ARG192 (1.864)
**9**	4	Pi-Sigma: TYR291, PHE251 Alkyl: LYS243, LEU283, LYS243 Pi-Alkyl: PHE251, PHE251, PHE251	TYR291 (3.528)

## Data Availability

Data are contained within the article.

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
