# Peer review of "Zingiber officinale Uncovered: Integrating Experimental and Computational Approaches to Antibacterial and Phytochemical Profiling"

_pharmaceuticals, 2024, doi:10.3390/ph17111551_

Round 1

Reviewer 1 Report (Previous Reviewer 1)

Comments and Suggestions for Authors

The article can be published in journal Pharmaceuticals.

Author Response

Reviewer report 1

The article can be published in journal Pharmaceuticals.

Response:

Thank you very much; we sincerely appreciate your professional peer-review.

Reviewer 2 Report (Previous Reviewer 2)

Comments and Suggestions for Authors

The manuscript is a compact study of the identified compound of ginger extract, which is good as a thesis work, but it is something that is already published in several sources the antibacterial of the ginger extract is done several times, so which is better here? 

The phrase on the abstract "The computational analyses showed that ginger compounds bind the 1JIJ and 2QZW with high binding affinities, which reach –9.5 kcal/mol" first what are those (1JIJ and 2QZW) and what that means? what are the author measuring? the authors know that but the reader not.  how different or similar are those results with the reported. comparison is necessary again what is new here? the docking in those targets or what?

CG-MS analysis there are several on ginger, again what is new here? comparison and what is new here?

This statement "In recent years, researchers have investigated Ginger's antibacterial properties [10]," is it true?  ginger antibacterial properties have been studied since last century see PMID: 2615416 and reports in there. 

Honestly, I do not see anything new here

Comments on the Quality of English Language

minor 

Author Response

Reviewer report 2

The manuscript is a compact study of the identified compound of ginger extract, which is good as a thesis work, but it is something that is already published in several sources the antibacterial of the ginger extract is done several times, so which is better here? 

Response:

Thank you for your valuable feedback. We understand that there is substantial research on the antibacterial properties of ginger. However, our study is distinct in its focus on the Zingiber officinale rhizome grown in Sudan, a region where such data is limited. The local climate and soil conditions can influence the phytochemical composition, potentially offering unique insights into its bioactivity. Additionally, our work combines detailed GC-MS analysis and computational studies, providing a more comprehensive understanding that could be beneficial for future pharmaceutical applications. We believe these aspects add value to the existing literature.

The phrase on the abstract "The computational analyses showed that ginger compounds bind the 1JIJ and 2QZW with high binding affinities, which reach –9.5 kcal/mol" first what are those (1JIJ and 2QZW) and what that means? what are the author measuring? the authors know that but the reader not.  how different or similar are those results with the reported. comparison is necessary again what is new here? the docking in those targets or what?

As recommended, we added explanations of 1JIJ and 2QZW in the abstract as follow: “TyrRS from S. aureus (1JIJ) and aspartic proteinase from C. albicans (2QZW)”.                     See lines 34-35

About the novelty of the current study, it has been previously reported that while the major compounds of the studied plants remain the same, the others usually vary and is mainly related to the region, climate, and soil condition. Furthermore, even that the compounds themselves did not change, the proportion may significantly vary. In this context, we also inserted the following sentences “It has been reported that while the compounds themselves did not change, their proportion may significantly vary from one region to another. In this context, various phytochemical composition of the same plant might result in different biological activity, which is mainly related to the interactions of the identified phytochemicals themselves with the targeted macromolecules. Hence, the outlined antimicrobial effect of the studied Ginger may be the result and/or the consequence of its phytochemical composition.” See Section 3.3.

CG-MS analysis there are several on ginger, again what is new here? comparison and what is new here?

See previous answer of the previous comment.

This statement "In recent years, researchers have investigated Ginger's antibacterial properties [10]," is it true?  ginger antibacterial properties have been studied since last century see PMID: 2615416 and reports in there. 

Reviewer 3 Report (New Reviewer)

Comments and Suggestions for Authors

The authors revised the manuscript in accordance with the comments made. The manuscript may be accepted for publication

Comments on the Quality of English Language

Average

Author Response

Reviewer report 3

The authors revised the manuscript in accordance with the comments made. The manuscript may be accepted for publication

Response:

Thank you very much; we sincerely appreciate your professional peer-review.

Reviewer 4 Report (New Reviewer)

Comments and Suggestions for Authors

The manuscript is significant but needs improvements in the following areas.

What is the novelty of this study as many works have been carried out on the antibacterial activity of Zingiber officinale? The novelty aspect should be mentioned in the introduction and abstract.

Provide original GC-MS chromatogram.

The references cited in the introduction are not sufficient.

Quantification of total phenols, total flavonoids, and total alkaloids is recommended.

Comments on the Quality of English Language

The manuscript need minor spell check and grammar check.

Author Response

Reviewer report 4

The manuscript is significant but needs improvements in the following areas.

What is the novelty of this study as many works have been carried out on the antibacterial activity of Zingiber officinale? The novelty aspect should be mentioned in the introduction and abstract.

Response:

Thank you so much, the novelty aspects are included now, as there is limited information on bioactivity of medicinal plants from Sudan, a poor country with political conflicts. Moreover, our study is distinct in its focus on the Zingiber officinale rhizome grown in Sudan, and such data are limited. The local climate and soil conditions can influence the phytochemical composition, potentially offering unique insights into its bioactivity. Additionally, our work combines detailed GC-MS analysis and computational studies, providing a more comprehensive understanding that could be beneficial for future pharmaceutical applications. We believe these aspects add value to the existing literature.

Provide original GC-MS chromatogram.

The references cited in the introduction are not sufficient.

Response:

Thank you, we have now enriched the introduction according to your recommendation

Quantification of total phenols, total flavonoids, and total alkaloids is recommended.

Response:

Thank you for your suggestion to quantify total phenols, flavonoids, and alkaloids. While we recognize the value of these analyses, our study primarily focuses on a detailed characterization of the phytochemical profile using GC-MS, which identified 22 chemical compounds. GC-MS analysis provides a thorough understanding of the specific bioactive compounds present in the extract, offering insights into their potential contributions to the observed antibacterial and antioxidant activities. This approach allows us to identify key molecules and understand their structural properties, which may be more informative than general quantification of phenolic or flavonoid content. We believe that this detailed chemical profiling is sufficient to support the aims of our study and to provide a solid foundation for exploring the therapeutic potential of Sudan-grown Zingiber officinale

Comments on the Quality of English Language.

The manuscript need minor spell check and grammar check.

Response:

Thank you. We have submitted the manuscript for proofreading to MDPI, where it was reviewed and refined by native speakers. We have also included the proofreading certificate for your reference.

Round 2

Reviewer 2 Report (Previous Reviewer 2)

Comments and Suggestions for Authors

The manuscript is well written and well presented, but the authors failed in do comparison with other zingiber officinale species in similar climate and their components and activity. I do not see any of those data. the answer to my questions were not good and adding for example: "The Sudanese rhizome contains many bioactive compounds, including flavonoids, terpenoids, and phenolic acids" means that the other studied zingiber rhizomes do not contain flavonoids and terpenoids?? so what other rhizomes those contains? I do not think that the manuscript gets to a level to publish in a high impact journal. 

Comments on the Quality of English Language

minor

Author Response

Review Report (Reviewer 2)

The manuscript is well written and well presented, but the authors failed in do comparison with other zingiber officinale species in similar climate and their components and activity. I do not see any of those data. the answer to my questions were not good and adding for example: "The Sudanese rhizome contains many bioactive compounds, including flavonoids, terpenoids, and phenolic acids" means that the other studied zingiber rhizomes do not contain flavonoids and terpenoids?? so what other rhizomes those contains? I do not think that the manuscript gets to a level to publish in a high impact journal. 

Response:

Dear Reviewer,

Thank you for your valuable feedback on our manuscript. We appreciate your recognition of the overall quality of the writing and presentation.

Regarding your concern about the comparison with other Zingiber officinale species grown in similar climates, we acknowledge that this aspect could strengthen our study. While we focused on the Sudanese ginger rhizome, we recognize that including data from other ginger cultivars cultivated in comparable climatic conditions (like Egypt for example) would provide a more comprehensive analysis.

Accordingly we added the following :

Among the phenolic chemicals found in ginger, studies have shown that zingerone, gingerols, and shogaols are the most common. Of these, gingerols are commonly found in greater abundance in fresh ginger, while shogaols are formed through heat treatment. Zingiberene, β-bisabolene, and sesquiphellandrene are common terpenoids that are part of another important group of bioactive compounds. These compounds have antimicrobial and anti-inflammatory effects in various varieties, including those from Nigeria, China, and India [44, 45]. Research conducted in different climates on these chemicals has shown that although all rhizomes of Z. officinale contain the same class of compounds, the amounts and bioactivity of these compounds might vary [44, 46].

By comparing the GC-MS results of Sudanese ginger with those from a published study by Elbestawy et al. (Elbestawy et al 2023) [47] on Egyptian ginger, our study identified a total of 22 compounds in Sudanese ginger rhizome, whereas the published study identified 17 distinct compounds. Both studies revealed similarities and differences in the chemical composition of ginger rhizome. Gingerol and zingiberene were identified as the primary compounds in both, albeit in slightly different concentrations. Additional shared compounds include α-terpineol and β-bisabolene. Both studies emphasize ginger's pharmacological potential, noting its anti-inflammatory, antimicrobial, and antioxidant effects. However, there are notable differences. The published study detected thymol at a concentration of 10.50%, a compound absent in our analysis. Thymol is known for its antimicrobial and antifungal properties, contributing an additional aspect to ginger’s medicinal profile. Conversely, our study identified unique compounds such as β-D-glucopyranose, 4-hydroxybenzoic acid, and exo-2,7,7-trimethylbicyclo[2.2.1]heptane-2-ol. Additionally, the published study identified shogaol and α-bergamotene, unique compounds with anti-inflammatory properties that further enhance ginger’s therapeutic potential. These differences highlight potential variations in ginger’s chemical composition, influenced by factors such as plant origin, extraction methods, and analytical conditions. We conclude that the constituents of ginger are significantly affected by the cultivation environment. Our observation is supported by a previous study on ginger from Indonesia and Japan, where GC-MS analysis showed clear differences in composition between the ginger cultivars [48].

  1. Elbestawy, M. K., El-Sherbiny, G. M., Moghannem, S. A., & Farghal, E. E. (2023). Antibacterial, Antibiofilm, and Anti-Inflammatory Activities of Ginger Extract against Helicobacter pylori. Microbiology Research14(3), 1124-1138.
  2. Nishidono, Y., Saifudin, A., Deevanhxay, P., & Tanaka, K. (2020). Metabolite profiling of ginger (Zingiber officinale Roscoe) using GC-MS and multivariate statistical analysis. Journal of the Asia-Japan Research Institute of Ritsumeikan University2, 1-14.

13/11/2025

Round 3

Reviewer 2 Report (Previous Reviewer 2)

Comments and Suggestions for Authors

Finally, the authors did some comparison.

This manuscript is a resubmission of an earlier submission. The following is a list of the peer review reports and author responses from that submission.

Round 1

Reviewer 1 Report

Comments and Suggestions for Authors

The authors did not provide a reference to the article on the antimicrobial effect of plant extracts (https://doi.org/10.59761/RCR5108).

Reviewer 2 Report

Comments and Suggestions for Authors

The manuscript deal with the phytochemical study and antibacterial study a well-known medicinal plant, the Sudanese ginger. The study seems well designed, still the language is not at the point of a high impact journal. The abstract is highlighting the use of ginger, which for me it is too much saying that: " lays the groundwork for future research and pharmaceutical exploration aimed at harnessing the beneficial properties of ginger rhizome for medicinal and therapeutic purposes, particularly its antimicrobial potential" It is a well-known medicinal plant and heavily use one for several illnesses and maladies. The use of several words in the text that are not usually use in writing scientific language make me suspicious of AI incorporate in the manuscript. which is not bad, but the authors must give to somebody with knowledge of scientific language. 

Several other antibacterial, phytochemical investigations had been done in the literature, but the authors do not compare those with the current, which is the groundbreaking difference, comparison of that it is important. What the Sudanese ginger has that no other has? 

The computational work even supports the data, but also it is already done in several other reports, so which is the difference?

The data of the current manuscript is not exciting, and it does not support a high impact journal as Pharmaceuticals, I suggest a lower impact journal after some modifications.

Comments on the Quality of English Language

some grammatical mistakes still are in there.

Reviewer 3 Report

Comments and Suggestions for Authors

Dear Authors, thank you for trying to address the medicinal plants available on the local market. However, the manuscript is unacceptable in its present form to be published anywhere, not only in Pharmaceuticals, for some substantial reasons. Details are provided below.

:: main issues ::

l.25

"this variety" => Introduce the name of this "variety" first. If it is not a "variety" in the botanical sense, use another term instead and rewrite the sentence.

l.33 and later

"M.B.C." vs. "MBC" => Use one form consistently. Similarly, for "MIC".

l.34, also in Keywords

"The phytochemical screening revealed the presence of various chemical constituents such as saponins, flavonoids, glycosides, alkaloids, steroids, and terpenoids and the absence of tannins"

=> Avoid populism. If you do not have serious proof for any of the mentioned compounds, do not report them. E.g., "saponins" - there is no evidence of saponins in genus Zingiber. If you found saponins in ginger, it would be a unique discovery, deserving of dissemination in a much higher-ranking journal. However, after reading the rest of your manuscript, I found no factual evidence of saponins. Using the unproven terms in the Abstract and the Keywords is a way to mislead the databasing engines.

=> Consider that saponin-rich "yellow ginger = Dioscorea zingiberensis C. H. Wright" has no botanical relevance to natural ginger (= Zingiber officinale Roscoe).

l.37

"with retention times ranging from 7.564 to 17.023 minutes"

=> Is it worthy to use the Abstract space for such details?

l.40

"ginger compounds bind the 1JIJ and 2QZW"

=> Before you use an abbreviation, introduce its whole meaning. BTW, what is the medical effect of such binding?

l.42

"that satisfactorily explain the in Vitro results" => Does it really satisfactorily explain the results?

l.109

"The present study aims to comprehensively analyze ginger rhizome's antibacterial, phytochemical, and GC-MS properties and explore its possible applications"

=> There is a problem with this aim: Why comprehensively analyze the mentioned properties of ginger if so much has already been done, and we know the basics in this field?

l.121

"pharmacological qualities of Sudanese medicinal plants"

=> Again, introducing and botanical defining your ginger variety first will give you a more substantial starting point to analyze it. Now it is just undefined Sudanese ginger (undefined in the meaning: Other researchers cannot repeat your results because they do not know precisely the object of your study).

l.202

"while at the lower concentration, it averaged 11.0±0.1 mm"

=> You have defined the higher concentration but not the lower one.

l.223/Table 1

=> Lack of column with positive reference (antibiotic and its amount per disc).

l.229

"A previous study suggested that the antibacterial properties of Ginger are linked to the presence of naphthalenamine, decanal, and alfa-copaene [27]"

=> Nice, but you have water extract, in which the concentration of lipid-soluble essential oil constituents is probably negligible (unless you prove it).

l.229 !

=> There is no evidence for compounds "naphthalenamine, decanal, and alfa-copaene" in source [27].

l.223/Table 1 vs. l.244/Table 2

=> Why was aqueous extract replaced with ethanol extract in the second case?

l.246-253 + Table 3 !

=> How was the presence of the listed metabolites assured? What is the level of uncertainty of your simple qualification "present/absent"?

=> To what does the "relatively more quantity" refer?

l.260 !

=> How did you find volatile disaccharide under described GCMS conditions?

l.276/Table 4 !

=> What is the base for identifying compounds listed in Table 4? RI? Standards?

=> Non-volatile compounds are reported. Stereochemically defined compounds are reported.

=> "Name" and "Formula" of some compounds do not agree (e.g., peak 16).

l.279+ !

=> The analysis is based on uncertain results from GCMS. Thus, unjustified.

=> Why do you explore kaempferol? You do not have serious proof of the presence of this molecule in your plant material, and you do not know the amount of this molecule in your plant material. => If you want to focus on kaempferol with computational modeling, do it in another manuscript.

RESULTS summary

You do not have any serious discussion with literature results on ginger antibacterial properties performed with disc diffusion test. Positive control is absent in your study. Similarly, there is no discussion with literature findings on metabolites (Tab.3) or volatile constituents (Fig.2/Tab.4). The GCMS study is unreliable because you do not provide any details on the identification strategy. The Computational Modeling part explores kaempferol, a molecule neither sufficiently identified in this study nor present in a serious amount in ginger.

l.323-329 / Discussion

"The present study aims to comprehensively analyze ginger rhizome's antibacterial, phytochemical, and GC-MS properties and explore its possible applications. This study provided a detailed insight into the various chemical constituents present in the ginger rhizome, their potential antibacterial effects, and possible therapeutic uses. The findings of this research will contribute to the global understanding of Ginger's health benefits and support its widespread use in various medical and culinary applications."

=> I don't see it this way. Based on my findings above, this study is unreliable and will cause unnecessary scientific mess.

l.342

=> There is no single compound like "gingerol", "shogaol" or "paradol". All these names are family names (e.g., you have [6]-gingerol and [8]-gingerol, numerous paradols, and so on). BTW, why the only one of them identified in your study, is "gingerol" (Table 4)?

l.354

=> The type of extract, its DER (drug-to-extract ratio) or DSR (drug-to-solvent ratio), and the amount applied per disc are needed to compare the inhibition zones. Citing literature's inhibition zones alone makes no sense. Your writing style results in a populism, suggesting to an uninformed reader that your and cited conditions for disc-diffusion assay were the same (e.g., your 10 mm can be compared to the literature 4.9 mm inhibition zone).

l.373-391 !

"The presence of phenols and tannins, alkaloids, flavonoids, terpenoids, sterols, cardiac glycosides, and saponins in the extracts"

=> OK, so cardiac glycosides are also welcome in the club? Do you understand the principles of phytochemical analysis methods at all?

"Plant-based natural compounds, such as alkaloids, have shown promise as preventive measures against chronic inflammation and neurodegenerative diseases"

=> How can you generalize about the properties of such a chemically complex and diverse (in terms of biological properties) group?

l.432

"the alkaloid gingerol (18%)"

=> OK, so now gingerol is an alkaloid?

l.479-482

=> Merck is a German chemical company.

=> No defined standard let me judge that no standards (for antibacterial, GC-MS, ... assays) were used.

l.486-487

=> When providing the precise latitude and longitude, use geographic directions, too.

l.495-500

=> In Table 1, you report the results obtained for the aqueous extract. Here is only the description of 80% methanol extract.

=> In this way, you obtain a thick extract with unknown water remains. Thus, other scientists cannot repeat your experiment. It is uncomfortable, but if you know the total amount of extract of your 100g of plant material, you can still express the amount of extract applied per disc as the amount of ginger rhizome equivalent. That will be more repeatable.

l.501-515

=> I am afraid that [62] is misleading. The methods described there are general, XX-century methods, that are not robust to various compounds giving false-positive reactions. The methods can be used as preliminary rather than indisputably confirming the identity of groups of compounds.

Let's see and comment what is given in [62]:

a. Determination of saponins

=> Different groups of compounds give the foam under such conditions. Other properties should be checked (e.g., the haemolytic activity of foaming fractions of your extract; amphiphilic properties observed during wisely conducted thin-layer chromatography (TLC); liquid chromatography with high-resolution mass spectrometry (LC-HRMS) and with fragmentation mass spectrometry (LC-MS/MS) methods are useful for saponin identification in analytical scale).

b. Determination of tannins

=> Such a test is valid for hydrolyzing tannins. Unfortunately, different free phenolics also react with FeCl3. Preliminary only; still valuable in combination with wisely conducted TLC.

c. Determination of alkaloids

=> Mayer's reagent reacts with numerous alkaloids, but also with other compounds. Preliminary only; still valuable in combination with wisely conducted TLC (however, Dragendorff reagent is better).

d. Determination of glycosides

=> Highly unspecific. Concentrated sulphuric acid is a strong dehydrating reagent, producing different colored molecules from one pure molecule. The resultant color depends on temperature, time, concentration, ... and most of all, the presence of secondary constituents. Thus, it is a test that is completely useless for extracts.

e. Determination of flavonoids

=> Reducing ketone, like in the chromone ring in flavonoid, results in double-bond bridges, which are sometimes colored, as [62] describes. However, this reaction is not addressed to flavonoids only. In ginger, you have aromatic ketones like gingerols. The resultant color depends on temperature, time, concentration, ... and most of all, the presence of secondary constituents. Thus, it is a rather useless test for extracts, except for experienced researchers (then: preliminary test).

=> Nothing about cardiac glycosides.

=> Again, if you know the total amount of extract obtained from 100g of your plant material, you can express the amount of extract diluted in stock as amount of ginger rhizome equivalent. That will be more repeatable.

l.680

"grown in the central part of Sudan (Gezira)"

=> You purchased it from the local market but do not know where it was grown.

Comments on the Quality of English Language

:: language issues ::

l.84

"researchers [...] explore Ginger's antibacterial, phytochemical, and GC-MS properties"

=> Avoid mixing biological and chemical properties.

l.97-101

"The rhizome contains many bioactive compounds"

=> Avoid leaving guesses. What do you mean by writing "many"? The unusual content/yield? The unusual diversity? Flavonoids in this set look weird because the common literature standard reports about 0.005-0.05% of total flavonoids in the ginger rhizome (calculated on dry mass).

"In particular, ginger flavonoids, such as quercetin and kaempferol, have been extensively studied for their potential to combat various diseases"

=> OK, but refer to their amounts in your plant material first. Reporting super-properties of compounds almost absent in plant material is highly misleading.

l.128

"it is still rarely used in medicine"

=> Imprecise. What is still rarely used in medicine?

l.199/200

=> A part of the manuscript text is missing.